# Involvement of G-Protein-Coupled Receptor 40 in the Inhibitory Effects of Docosahexaenoic Acid on SREBP1-Mediated Lipogenic Enzyme Expression in Primary Hepatocytes

**DOI:** 10.3390/ijms20112625

**Published:** 2019-05-28

**Authors:** Seungtae On, Hyun Young Kim, Hyo Seon Kim, Jeongwoo Park, Keon Wook Kang

**Affiliations:** College of Pharmacy and Research Institute of Pharmaceutical Sciences, Seoul National University, Seoul 08826, Korea; seungtae.on@gmail.com (S.O.); hens93@snu.ac.kr (H.Y.K.); hyoseonkim@snu.ac.kr (H.S.K.); jwpark0110@snu.ac.kr (J.P.)

**Keywords:** GPR40, GPR120, DHA, omega-3 fatty acid, SREBP-1, hepatocytes

## Abstract

Nonalcoholic fatty liver disease is a frequent liver malady, which can progress to cirrhosis, the end-stage liver disease if proper treatment is not applied. Omega-3 fatty acids, such as docosahexaenoic acid (DHA) and eicosapentaenoic acid, have been clinically proven to lower serum triglyceride levels. Various physiological activities of omega-3 fatty acids are due to their agonistic actions on G-protein-coupled receptor 40 (GPR40) and GPR120. Lipid droplets (LD) accumulation in hepatocytes confirmed that DHA treatment reduced the number of larger ( >10 μm^2^) LDs, as well as the total area of LDs. Moreover, DHA lowered protein and mRNA expression levels of lipogenic enzymes such as fatty acid synthase (FAS), acetyl-CoA carboxylase and stearoyl-CoA desaturase-1 (SCD-1) in primary hepatocytes incubated with liver X receptor (LXR) agonist T0901317 or high glucose and insulin. DHA also decreased protein expression of nuclear and precursor sterol response-element binding protein (SREBP)-1, a key lipogenesis transcription factor. We further found that exposure of murine primary hepatocytes to DHA for 12 h increased GPR40 and GPR120 mRNA levels. Specific agonists (Compound A for GPR120 and AMG-1638 for GPR40), hepatocytes from GPR120 knock-out mice and GPR40 selective antagonist (GW1100) were used to assess whether DHA’s antilipogenic effects are mediated through GPR120 or GPR40. Compound A did not decrease SREBP-1 and FAS protein expression in hepatocytes exposed to T0901317 or high glucose with insulin. Moreover, DHA downregulated lipogenesis enzyme expression in GPR120-null hepatocytes. In contrast, AMG-1638 lowered SREBP-1 and SCD-1 protein levels. Additionally, GW1100, a GPR40 antagonist, reversed the antilipogenic effects of DHA. Collectively, our data demonstrate that DHA downregulates the expression SREBP-1-mediated lipogenic enzymes via GPR40 in primary hepatocytes.

## 1. Introduction

Nonalcoholic fatty liver disease (NAFLD) affects 80–100 million people in the U.S. alone [1]. Most patients in the U.S., who visit primary care providers for routine checkup have elevated levels of alanine aminotransferase (ALT) and aspartate aminotransferase (AST), which are markers of liver damage [2]. Among those with elevated AST and ALT levels, 33 percent are eventually diagnosed with NAFLD [2]. This pattern is not limited to the Western world. In Asia, 25 percent of individuals aged >18 years are diagnosed with NAFLD [3]. Even though this widespread disease has no specific signs or symptoms, it can progress to nonalcoholic steatohepatitis (NASH) and eventually lead to cirrhosis [4]. Although treatment options for NAFLD are limited, thiazolidinediones and vitamin E have been proven to effectively manage symptoms [5].

Omega-3 fatty acid provides an energy source as nutrient, but also act as a remedy for serum hyperlipidemia. Commercially available fish oil supplements are one of the most popular over-the-counter medications for the management of elevated serum triglyceride (TG) levels. In the U.S. alone, about 18.8 million adults are taking fish oil supplements [6]. Omega-3-acid ethyl esters, such as Omacor^®^ and Lovaza^®^, are widely used as prescription medications for dyslipidemia [7]. Omega-3 fatty acid supplementation decreases liver fat and reduces hepatic steatosis, as well as having beneficial effects on most cardiometabolic risk factors. According to clinical studies, dietary supplementation with omega-3 is efficacious for NAFLD management [8]. However, the mechanism of its TG-lowering effect is not fully understood.

Omega-3 fatty acids, such as docosahexaenoic acid (DHA) and eicosapentaenoic acid (EPA), have been recognized as endogenous ligands for G-protein-coupled receptor 40 (GPR40) and G-protein-coupled receptor 120 (GPR120) [9,10]. In animal models, the administration of selective agonists for GPR40 or GPR120 improves biochemical and pathological indices of hepatic steatosis [11,12]. Nevertheless, the proposed direct effects of omega-3 fatty acids on the G-protein-coupled receptor (GPCR) in hepatocytes are largely uncharacterized and controversial. In the present study, we assessed the effect of DHA on hepatic lipogenesis and attempted to clarify the roles of GPR40 and GPR120 in antilipogenesis effect of DHA in mouse primary hepatocytes.

## 2. Results

### 2.1. Downregulation of Sterol Regulatory Element Binding Protein (SREBP)-1-Dependent Lipogenic Enzyme Expression by DHA in Primary Hepatocytes

Public microarray data comparing control mice with mice receiving omega-3 fish oil for 2 weeks were analyzed to assess whether omega-3 fatty acids functionally benefit liver (GSE32706). Gene ontology analyses revealed that the genes, including those involved with lipid metabolic processes and cholesterol biosynthetic processes, were mainly altered for similar functional annotations in the liver tissues of fish-oil-fed mice (Appendix A). In addition, expression of genes involved in lipid metabolism and fatty acid synthesis was repressed in the fish-oil-fed group (Appendix A). Moreover, fish oil diet led to a significant reduction in the expression of several genes, including *srebf1*, *acacb*, *fasn*, and *scd1* which are required to control hepatic TG synthesis (Appendix A).

The SREBP1 transcription factor is a critical regulator of fatty acid homeostasis in hepatocytes. Activation of SREBP1 controls the expression of a range of lipogenic enzymes, such as fatty acid synthase (FAS), acetyl-coenzyme A carboxylase (ACC), and stearoyl-CoA desaturase-1 (SCD-1) [13]. Hence, we analyzed the effects of DHA and EPA (300 μM, each) on the expression of SREBP1 and its downstream lipogenesis enzymes. The levels of FAS, ACC, and SCD-1 were diminished in mouse primary hepatocytes treated with DHA or EPA for 12 h, and the degree of inhibition was more prominent in DHA-treated hepatocytes (Figure 1A). The protein level of the precursor form of SREBP-1 (preSREBP-1) was decreased in primary hepatocytes in response to DHA (Figure 1B). To gain further insight into the antilipogenic properties of DHA, primary hepatocytes were treated with liver X receptor (LXR) agonist T0901317 (T090) to stimulate LXR-dependent SREBP-1 activation. T090-induced protein expression of nuclear SREBP-1 and SCD-1 were decreased significantly following pretreatment of hepatocytes with 300 μM DHA for 12 h (Figure 1C). Although the levels of preSREBP-1, FAS, and ACC were marginally elevated by T090, all the protein expression decreased in response to the DHA treatment (Figure 1D and Appendix A). Primary hepatocytes were exposed to high glucose with insulin condition to confirm the antilipogenic properties of DHA in metabolic dysfunction. As expected, the amounts of lipogenic proteins involved in TG synthesis, such as preSREBP-1, nSREBP-1, FAS, ACC, and SCD-1, increased under the high glucose with insulin condition, and the enhanced levels were reversed by DHA treatment (Figure 1E and Appendix A). These data support the notion that DHA ameliorates the SREBP-1-mediated lipogenesis enzyme expression caused by an LXR agonist or the high glucose with insulin condition.

### 2.2. DHA-Induced Reduction of Total Area of Lipid Droplets in Hepatocytes

Hepatic steatosis is defined as abnormal retention of lipid droplets (LD) in hepatocytes, which reflects the dysregulation of TG fat [14]. We explored the effect of DHA on LD formation in hepatocytes. BODIPY^®^ staining showed a greater increase in LDs in hepatocytes incubated under the high glucose and insulin condition, but not with T090 (Figure 2A,B). Interestingly, no difference was detected in the number of LDs per cell between hepatocytes treated with high glucose and insulin and those cotreated with DHA (Figure 2C). Nevertheless, the number of LDs > 10 μm^2^, representing pathological lipid accumulation, as well as the total area of LDs per cell decreased in response to DHA (Figure 2D,E). One study showed that the decrease in the size of LDs is due mainly to the lower activity of SCD-1 [15]. The immunoblot results showed a dramatic decrease in SCD-1 with DHA treatment (Figure 1C,D), suggesting that DHA mainly targets enlargement of LDs in hepatocytes.

### 2.3. Limited Role of GPR 120 in Antilipogenic Effects of DHA in Hepatocytes

Long-chain fatty acids, such as DHA, act as endogenous ligands on GPR120 and GPR40 and regulates metabolic and inflammatory homeostasis in adipocytes and macrophages [10,16]. To investigate whether GPR40 and GPR120 are present in hepatocytes, mRNA levels were determined in HepG2 human hepatoma cell line and mouse primary hepatocytes. mRNA levels of GPR40 and GPR120 were detectable in HepG2 and mouse primary hepatocytes (Appendix A and Figure 3A). Moreover, exposure to 300 μM DHA for 12 h increased the GPR40 and GPR120 mRNA levels (Appendix A and Figure 3A). Plasma membrane localization of GPR40 and GPR120 in primary hepatocyte was confirmed by immunocytochemistry (Figure 3B). These data demonstrate that GPR40 and GPR120 are expressed in hepatocytes and HepG2 cells are upregulated after DHA exposure. Compound A (CpdA), a specific GPR120 agonist [11], was used to clarify the function of GPR120 on the antilipogenic effects of DHA. Although the basal SCD-1 level was higher in HepG2 cells [17], CpdA concentration-dependently inhibited SCD-1 expression in T090-treated HepG2 cells and primary hepatocytes (Figure 3C and Appendix A). However, unlike DHA, CpdA up to 10 μM did not reduce the expression of preSREBP-1 in primary hepatocytes treated with T090 (Figure 3C). Because series of studies proposed direct interaction between LXR and SCD-1 [18,19], we presumed that downregulation of SCD-1 by CpdA may not be dependent on SREBP-1 pathway. Under the high glucose with insulin condition, CpdA did not reduce the protein level of preSREBP-1, nSREBP-1, and FAS in hepatocytes (Figure 3D and Appendix A). To confirm these findings, we used age-matched GPR120 knock-out (KO) mice. A pathological examination was performed in hematoxylin and eosin (H&E) stained liver tissues from 8-weeks-old GPR120 wild-type and KO mice to rule out preexisting histological conditions (Appendix AD). The decreased lipogenic enzyme expressions by DHA in primary hepatocytes incubated with T090 or high glucose and insulin were not reversed by GPR120 deficiency (Figure 3E,F and Appendix A). These data imply that the inhibitory effects of DHA on the expression of SREBP-1-mediated lipogenic enzymes are minimally associated with GPR120 signaling.

### 2.4. Involvement of GPR40 in Antilipogenesis Effect of DHA

We tested the potential role of GPR40 in antilipogenic effects of DHA in hepatocytes. AMG-1638 is a well-characterized GPR40 agonist in murine species and has shown a specific full agonistic activity compared with other candidates [20]. The enhanced expression of preSREBP-1 and SCD-1 caused by T090 was significantly diminished in hepatocytes incubated with 3 μM AMG-1638 (Figure 4A). Moreover, 3 μM AMG-1638 reduced the protein expression of FAS and SCD-1 as well as preSREBP-1 and nSREBP-1, under the high glucose and insulin condition (Figure 4B). We further assessed the effect of the GPR40 antagonist GW1100 to confirm whether the antilipogenic properties of DHA were related with GPR40 activation. When hepatocytes were co-incubated with both GW1100 and DHA, the inhibitory effects of DHA on the expression of preSREBP-1 and SCD-1 were almost completely reversed by GW1100 in T090-exposed hepatocytes (Figure 4C). Similar results were obtained for GPR120 KO hepatocytes (Figure 4D). Hence, downregulation of SREBP-1-mediated expression of lipogenic enzymes by DHA was under the control of GPR40 in hepatocytes.

## 3. Discussion

DHA and EPA are omega-3 fatty acids and known as ligands for GPR40 and GPR120. Although they both have antilipogenic properties, DHA is associated with a greater reduction in serum TG levels compared with EPA [21]. In addition, DHA has a lipid control benefit over EPA, as it increases the serum level of high-density lipoprotein (HDL) [22]. Many studies have explored the potentially beneficial effects of GPR40 and GPR120 in various cells and organs. However, possibly due to limited expression of GPR40 and GPR120 in hepatocytes compared with other cell types [23,24], few studies have clearly identified the receptor involved in the antilipogenic effects of omega-3 fatty acids in the liver. In this study, we investigated the effects of DHA on lipogenic enzyme expression in primary hepatocytes and sought to clarify the related lipid-sensing GPCR exerting its antilipogenesis effect.

GPCRs are usually downregulated to eschew repercussive effects by persistent stimulation with agonists. Nobili et al. reported that patients administered DHA develop increased levels of GPR120 in hepatocytes [25]. This result is consistent with our finding that GPR120 and GPR40 mRNA levels are increased in mouse primary hepatocytes treated with DHA. Although regulatory mechanisms underlying the expression of long-chain fatty-acid-sensing GPCRs are not fully understood, Abaraviciene et al. showed that the GPR40 protein level is increased in response to 100- and 1000-μM palmitate exposures in rat islet [26]. However, GPR40 mRNA expression was enhanced only by 100 μM palmitate treatment. Therefore, we expect that both transcriptional and post-translational regulation play a role in the enhanced expression of GPR40 and GPR120 after DHA exposure. Moreover, immunohistochemical analyses using GPR40 and GPR120 antibodies confirmed that both GPR40 and GPR120 were present in mouse primary hepatocytes. Although GPR40 and GPR120 have different molecular structures, they share long chain fatty acids as their endogenous ligands, and the identification of highly specific ligands to discriminate the receptors is difficult [11]. Before CpdA was developed by Merck, TUG-891 was the most selective compound for GPR120. Unfortunately, TUG-891 loses its GPR120 selectivity in murine species [27]. Thus, in this study, CpdA was used to target GPR120 in mouse primary hepatocytes. We found that CpdA did not lower the protein expression of SREBP-1-dependent lipogenic enzymes stimulated by the high glucose with insulin milieu. Moreover, DHA reduced protein levels of preSREBP-1 and FAS in primary GPR120 KO hepatocytes exposed to T090 or high glucose and insulin. From this finding, we suggest that GPR120 is not mainly involved in the antilipogenic effects of DHA in hepatocytes. Contrary to our findings, Kang et al. reported that DHA and TUG-891 decreased LXR-mediated lipogenic protein expression in HepG2 and Hep3B hepatoma cell lines, as well as mouse primary hepatocytes, and further revealed that GPR120 is involved in the antilipogenic effect of DHA [28] .This discrepancy seems to be mainly to differences in the concentration range of DHA in cell-based analyses. In our experimental conditions, protein expression of preSREBP-1 and a series of lipogenic enzymes were suppressed by 300 μM DHA treatment. In contrast, others have reported that 10–30 μM DHA efficiently inhibited lipid accumulation by downregulating the SREBP-1 pathway in hepatoma cell lines and primary hepatocytes. DHA seems to have a potent binding affinity to GPR120. A DHA concentration < 1 μM evoked internalization of GPR120 in HEK293 cells stably expressing GPR120-enhanced green fluorescent protein (EGFP) [24]. Hence, relatively low concentrations of DHA may preferentially stimulate the GPR120 signal, whereas higher concentration act on GPR40. In fact, 10–20 μM DHA induces calcium influx in Chinese Hamster Ovary (CHO) cells expressing GPR40 [29].

Because the Gα_q/11_–coupled receptor GPR40 is expressed mainly in pancreatic β-cells, most of previous studies focused on its potential ability to stimulate insulin secretion. Because GPR40 activation stimulates insulin secretion only in the presence of elevated glucose levels [9,30,31], it has become an attractive potential therapeutic target for glucose homeostasis with little to no hypoglycemic risk. GPR40 is negligibly expressed in the liver [16]. Thus, the functional roles of GPR40 in hepatocytes have received less attention. In this study, we used GPR40 full agonist, AMG-1638, to assess whether GPR40 plays a functional role in hepatocytes. The enhanced lipogenic enzyme levels caused by LXR agonist or high glucose with insulin milieu were diminished by AMG-1638 in primary hepatocytes. Furthermore, GPR40 antagonist GW1100 efficiently abrogated the antilipogenic effects of DHA. These cell-based analyses using specific pharmacological tools suggest that receptor signaling plays an antilipogenic role in primary hepatocytes. Li et al. reported that the GPR40/120 agonist GW9508 improves hepatic steatosis in mice fed a high-cholesterol diet and inhibits LXR ligand-induced expression of lipogenic enzymes in HepG2 cells [12]. They further found that the antilipogenic activity of GW9508 is significantly reversed by GPR40 siRNA, suggesting a pivotal role of GPR40 in the regulation of hepatic lipid accumulation. These data are consistent with our findings.

Taken together, our findings may help to unravel how DHA alleviates fatty acid accumulation in hepatocytes. Furthermore, these findings support the notion of using specific GPCR agonists as an add-on therapy to manage metabolic syndrome and suggest that GPR40 merits further investigation as an adjuvant therapy for NAFLD.

## 4. Materials and Methods

### 4.1. Reagent and Antibodies

Antibody recognizing precursor sterol regulatory element binding protein-1 (SREBP-1) was obtained from Santa Cruz Biotechnology (Dallas, TX, USA). Anti-fatty acid synthase (FAS) and antinuclear form of SREBP-1 antibodies were supplied by BD Biosciences (Franklin Lakes, NJ). Stearoyl-CoA desaturase-1 (SCD1), phospho-AMP-activated protein kinase (p-AMPK), and acetyl-CoA carboxylase (ACC) antibodies were obtained from Cell Signaling Technology (Beverly, MA). Anti-glyceraldehyde 3-phosphate dehydrogenase (GAPDH) antibody and T0901317 (T090) were supplied by Calbiochem (San Diego, CA, USA). Anti-β actin antibody, insulin, and glucose were purchased from Sigma-Aldrich (St. Louis, MO). Horseradish peroxidase-conjugated donkey anti-rabbit IgG, and alkaline phosphatase-conjugated donkey anti-mouse IgG were obtained from Jackson Immunoresearch Laboratories (West Grove, PA, USA). Eicosapentaenoic acid (EPA), docosahexaenoic acids (DHA), and compound A (CpdA) were supplied by Cayman Chemical (Ann Arbor, MI, USA). GW1100 and AMG-1638 were kindly donated from LG Chem Ltd. (Seoul, South Korea).

### 4.2. Cell Culture

Human hepatoma cell line HepG2 was obtained from American Type Culture Collection (ATCC, Manassas, VA, USA). HepG2 cells were grown in low-glucose Dulbecco’s modified Eagle’s medium (DMEM) with 10% fetal bovine serum (FBS, Gibco, Thermo Fisher Scientific, Waltham, MA, USA) with 50 U/mL penicillin and 50 μg/mL streptomycin. All cell lines were maintained at 37 °C in a humidified incubator with 5% CO_2_.

### 4.3. Animals

GPR120 knock-out (KO) C57BL/6 mice were kindly donated from LG Chem Ltd. The mice were housed in a pathogen-free animal facility under a standard 12 h light/dark cycle. Animal experiments were conducted under the guidelines of the Institutional Animal Use and Care Committee at Seoul National University (SNU-160512-11-1, 12 May, 2016).

### 4.4. Isolation of Primary Hepatocytes

After anesthetizing 8-weeks-old C57BL/6 mice (DBL, Eumseong, Korea) or GPR120 KO mice with zoletil^®^ and rompun^®^, 24G catheter was cannulated to liver portal vein. After perfusion with Hank’s Balanced Salt solution medium (Life Technologies, Grand Island, NY, USA) supplemented with 0.5 mM ethylene glycol tetraacetic acid and 25 mM 4-(2-hydroxyethyl)-1-piperazineethanesulfonic acid (HEPES), liver tissue was digested with low-glucose DMEM containing 1% penicillin/streptomycin, 15 mM HEPES and 1 mg/mL collagenase from *Clostridium histolyticum *(Sigma-Aldrich, St.Louis, MO, USA). After tearing digested liver tissue, the isolated cells were washed three times with high-glucose DMEM supplemented with 10% FBS, 1% penicillin/streptomycin, 15 mM HEPES, 10 nM dexamethasone. Other cells except primary hepatocytes were all removed by centrifugation.

4.5. mRNA Isolation and Real-Time Quantitative Polymerase Chain Reaction (qPCR)

After washing with sterile phosphate-buffered saline (PBS), total RNA was isolated using TRIzol reagent (Life Technologies, Grand Island, NY). mRNA was reverse transcribed to cDNA using Maxime RT Premix (iNtRON, Seongnam, South Korea). Amplified cDNA was analyzed by Bio-Rad CFX Manager™ Software (Bio-Rad, Hercules, CA, USA) using iTaq Universal SYBR Green Supermix (Bio-Rad) and SYBR Select Master Mix (Life Technologies). Primer sequences used in experiments were:

5′-CTGTGCAGGAATGAGTGGAAG-3′ (mouse GPR120-forward)

5′-CTGATGGAGGGTACTGGAAATG-3′ (mouse GPR120-reverse)

5′-CTGTGCAGGAATGAGTGGAAG-3′ (human GPR120-forward)

5′-CTGATGGAGGGTACTGGAAATG-3′ (human GPR120-reverse)

5′-TGGCGCGCCAGCCTGG-3′ (mouse GPR120 KO-forward)

5′-CCATATGAAAGCCAGCAGTGCC-3′ (mouse GPR120 KO-reverse)

5′-CGCCGTCGGCGCCGTG-3′ (mouse GPR120 WT-forward)

5′-CCATATGAAAGCCAGCAGTGCC-3′ (mouse GPR120 WT-reverse)

5′-TTTCATAAACCCGGACCTAGGA-3′ (mouse GPR40-forward)

5′-CCAGTGACCAGTGGGTTGAGT-3′ (mouse GPR40-reverse)

5′- GCCCACTTCTTCCCACTCTA-3′ (human GPR40-forward)

5′- AGACCCAGGTGACACAGGAC -3′ (human GPR40-reverse)

5′-GTAACCCGTTGAACCCCATT-3′ (mouse and human 18s rRNA protein-forward)

5′-CCATCCAATCGGTAGTAGCG-3′ (mouse and human 18s rRNA protein-reverse)

### 4.6. Immunoblot Analysis

After washing the cultured cells with PBS, cells were lysed in lysis buffer containing 20 mM TrisHCl (pH 7.5), 1% Triton X-100, 137 mM sodium chloride, 10% glycerol, 2 mM EDTA, 1 mM sodium orthovanadate, 25 mM β-glycerophosphate, 2 mM sodium pyrophosphate, 1 mM phenyl methyl sulfonyl fluoride, and 1 μg/mL leupeptin. Cells were then incubated in ice for 1 h. The cell lysates were centrifuged at 10,000 *g* for 20 min to remove debris, and the protein samples were loaded on 8–15% SDS-PAGE gel and transferred to nitrocellulose membrane (GE healthcare Life Sciences, Chalfont, Buckinghamshire, UK). Membranes were incubated for 1 h with 5% skim milk (BD Bioscience, San Jose, CA, USA) and reacted with primary antibodies overnight at 4 °C. The membranes were washed and incubated with a second antibody for 1 h at room temperature. Protein expression was visualized with LAS3000-mini (Fujifilm, Tokyo, Japan) using enhanced chemiluminescence (ECL) system reagent (EMD Millipore, Billerica, MA, USA). Densitometric analysis was performed by using Multiguage software (Fujifilm, Tokyo, Japan) and Image J 1.46r.

### 4.7. Hematoxylin and Eosin Staining

The left lateral lobe of the liver was sliced, and tissue slices were fixed in 10% buffered-neutral formalin. The liver slices were stained with hematoxylin and eosin (H&E).

### 4.8. Detection of Lipid Droplets (LDs) by Confocal Fluorescence Scanning Microscopy

For BODIPY^®^ 493/503 staining (Thermo Fisher Scientific, Waltham, MA, USA), the 488 nm laser was used and signals were collected with a long pass 505 nm filter. Quantification of LD number and size was performed with MetaMorph, version 7.5 (Molecular Devices, San Jose, CA, USA).

### 4.9. Statistical Analysis

The analyses were performed with IBM^®^ SPSS^®^ Statistics 23 (IBM SPSS Statistics, IBM Corporation). The significance level was set at *p* value < 0.05 for all comparisons. Although the size of sample is small, Shapiro–Wilk test showed normality of the data and one-way analysis of variance (ANOVA) was used for multiple comparisons. Tukey’s or Dunnett’s tests were used as post hoc analysis methods. Statistical analysis for GSE data was performed by using Student’s *t*-test and Benjamini–Hochberg test.

## Figures and Tables

**Figure 1 ijms-20-02625-f001:**
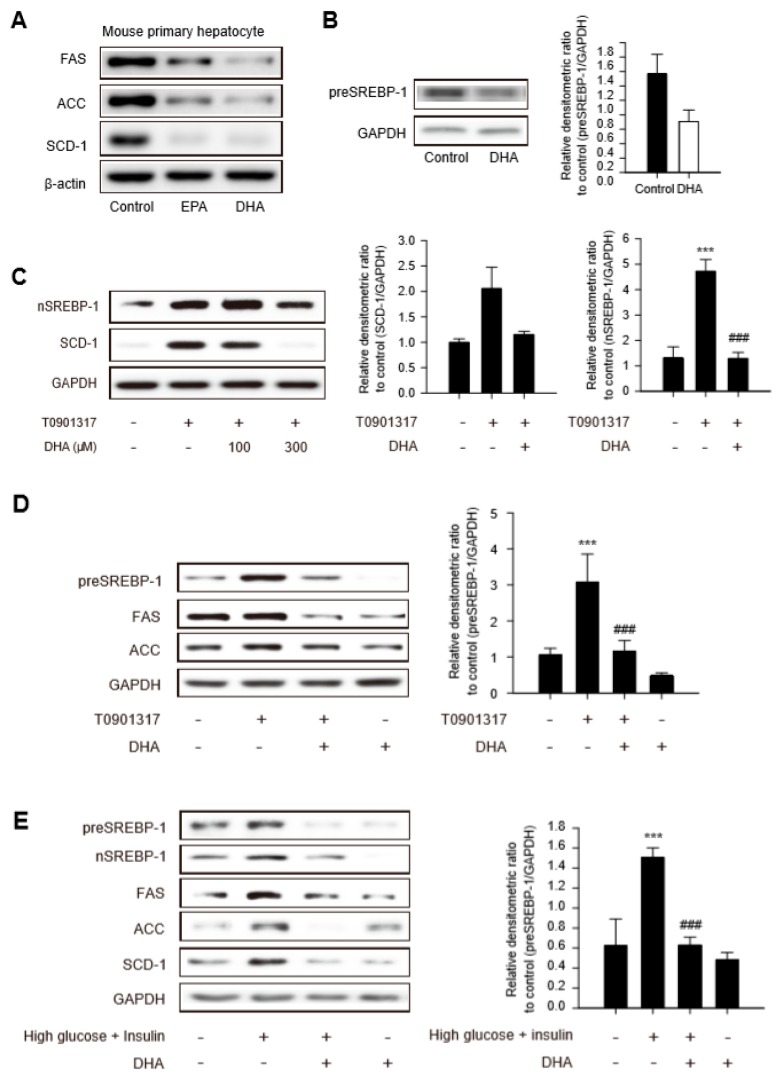
Decrease in lipogenesis enzymes after docosahexaenoic acid (DHA) treatment. (**A**,**B**) The protein expression level of fatty acid synthase (FAS), acetyl-coenzyme A carboxylase (ACC), stearoyl-CoA desaturase-1 (SCD-1) and precursor form of SREBP-1 (preSREBP-1). Mouse primary hepatocytes were treated with DHA 300 μM or eicosapentaenoic acid (EPA) 300 μM for 12 h. Data represent means ± SD (*n* = 5). GAPDH: Glyceraldehyde 3-phosphate dehydrogenase. (**C**) In order to screen an optimal DHA concentration that exerts potent antilipogenic effects, primary hepatocytes were treated with DHA 100 μM and 300 μM for 12 h followed by T0901317 (T090), liver X receptor (LXR) agonist for additional 12 h. Data represent means ± SD (*n* = 5), *** *p* < 0.005 compared with the control group; ^###^
*p* < 0.005 compared with the T090-treated group (**D**) Effects of DHA on the expression of lipogenic proteins stimulated by LXR agonist. Primary hepatocytes were treated with DHA 300 μM for 12 h followed by T090 for additional 12 h. Data represent means ± SD (*n* = 5), *** *p* < 0.005 compared with the control group; ^###^
*p* < 0.005 compared with the T090-treated group (**E**) Effects of DHA on the expression of lipogenic proteins in high glucose with insulin condition. Primary hepatocytes were treated with DHA 300 μM for 12 h followed by 30 mM high-glucose medium for 30 min and further incubation with 200 nM insulin for 24 h. Data represent means ± SD (*n* = 3), *** *p* < 0.005 compared with the control group; ^###^
*p* < 0.005 compared with the high glucose and insulin group.

**Figure 2 ijms-20-02625-f002:**
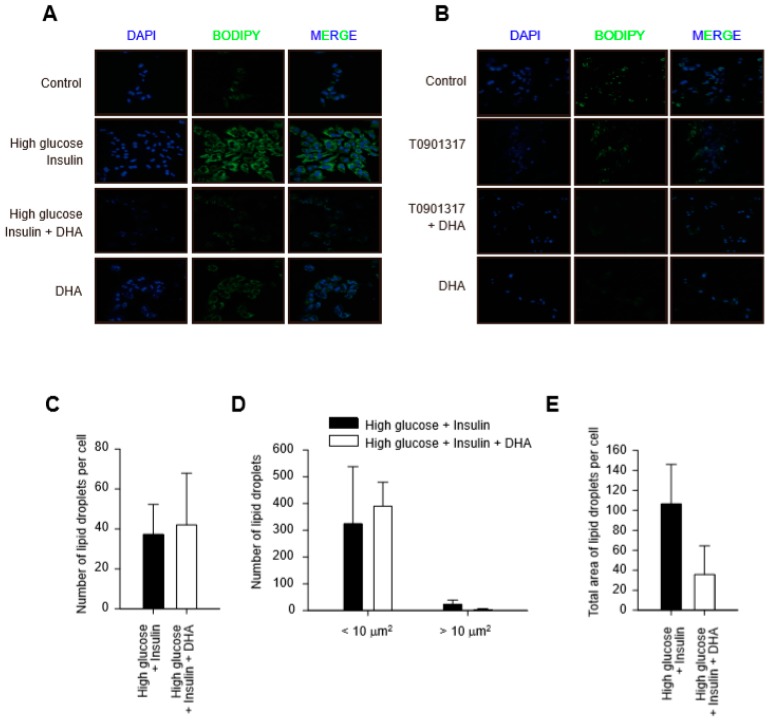
Morphological changes in lipid droplets (LD) after DHA treatment. (**A**,**B**) LD formation in hepatocytes by T090 treatment or high glucose with insulin condition. Mouse primary hepatocytes were treated with DHA for 12 h followed by treatment of T090 for 12 h or incubation with 30 mM high glucose for 30 min and further incubation with 200 nM insulin for 24 h. LDs were stained with BODIPY^®^ 493/503 and visualized by confocal microscopy. (**C**) Images of 16 random cells from each slide were captured and analyzed by MetaMorph to quantify number of LDs per cell. (**D**) Numbers of LDs with areas ≤10 µm^2^ or >10 µm^2^ were analyzed. (**E**) Total area of LDs per cell were measured and presented as means ± SEM.

**Figure 3 ijms-20-02625-f003:**
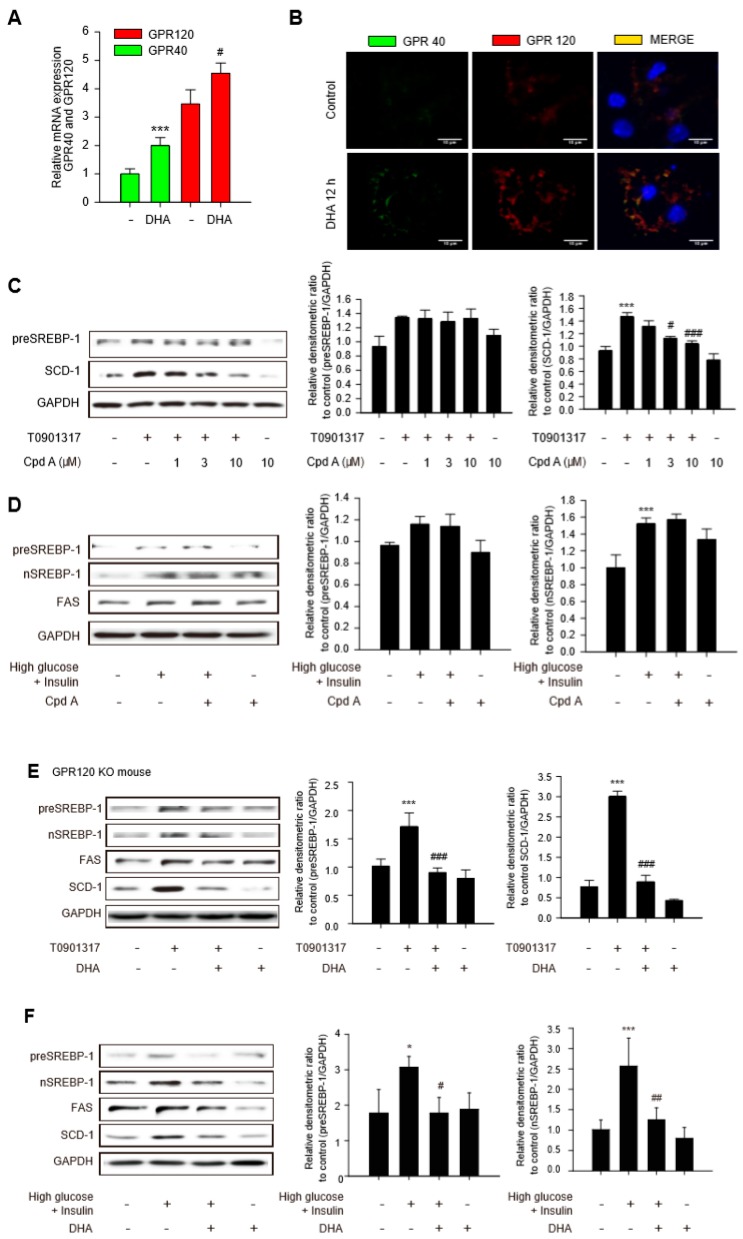
Role of GPR120 in expression of lipogenesis enzymes in hepatocytes. (**A**) Relative mRNA expressions of GPR40 and GPR120 levels were determined by real-time qPCR analyses in mouse primary hepatocytes. Data represent means ± SD (*n* = 3). *** *p* < 0.005 compared with the GPR40 level of the control group; ^#^
*p* < 0.05 compared with the GPR120 level of the control group. (**B**) Immunocytochemistry of GPR40 and GPR120 in primary hepatocytes. Immunofluorescent images of GPR40 and GPR120 at their basal levels and after DHA treatment are shown. (**C**) Primary hepatocytes were treated with compound A (CpdA) (1–10 μM) for 12 h followed by T090, LXR agonist for additional 12 h. Data represent means ± SD (*n* = 5), *** *p* < 0.005 compared with the control group; ^#^
*p* < 0.05, ^###^
*p* < 0.005 compared with the T090-treated group. (**D**) Effects of CpdA on the expression of preSREBP1, nSREBP1, and FAS in primary hepatocytes. Primary hepatocytes were treated with CpdA 10 μM for 12 h followed by 30 mM high-glucose medium for 30 min and further incubation with 200 nM insulin for 24 h. Data represent means ± SD (*n* = 5), *** *p* < 0.005 compared with the control group. (**E**) Effects of DHA on the expression of preSREBP1, nSREBP1, FAS, and SCD-1 in GPR120 knock-out (KO) primary hepatocytes stimulated with T090. GPR120 KO hepatocytes were treated with DHA for 12 h followed by T090 for additional 12 h. Data represent means ± SD (*n* = 3), *** *p* < 0.005 compared with the control group; ^#^
*p* < 0.05, ^###^
*p* < 0.005 compared with the T090-treated group. (**F**) Effects of DHA on the expression of preSREBP-1, nSREBP-1, FAS, and SCD-1 in GPR120 KO primary hepatocytes stimulated with high glucose and insulin. GPR120 KO primary hepatocytes were treated with DHA 300 μM for 12 h followed by 30 mM high-glucose medium for 30 min and further incubation with 200 nM insulin for 24 h. Data represent means ± SD (*n* = 3), * *p* < 0.05, *** *p* < 0.005 compared with the control group; ^#^
*p* < 0.05, ^##^
*p* < 0.01 compared with the high glucose and insulin group.

**Figure 4 ijms-20-02625-f004:**
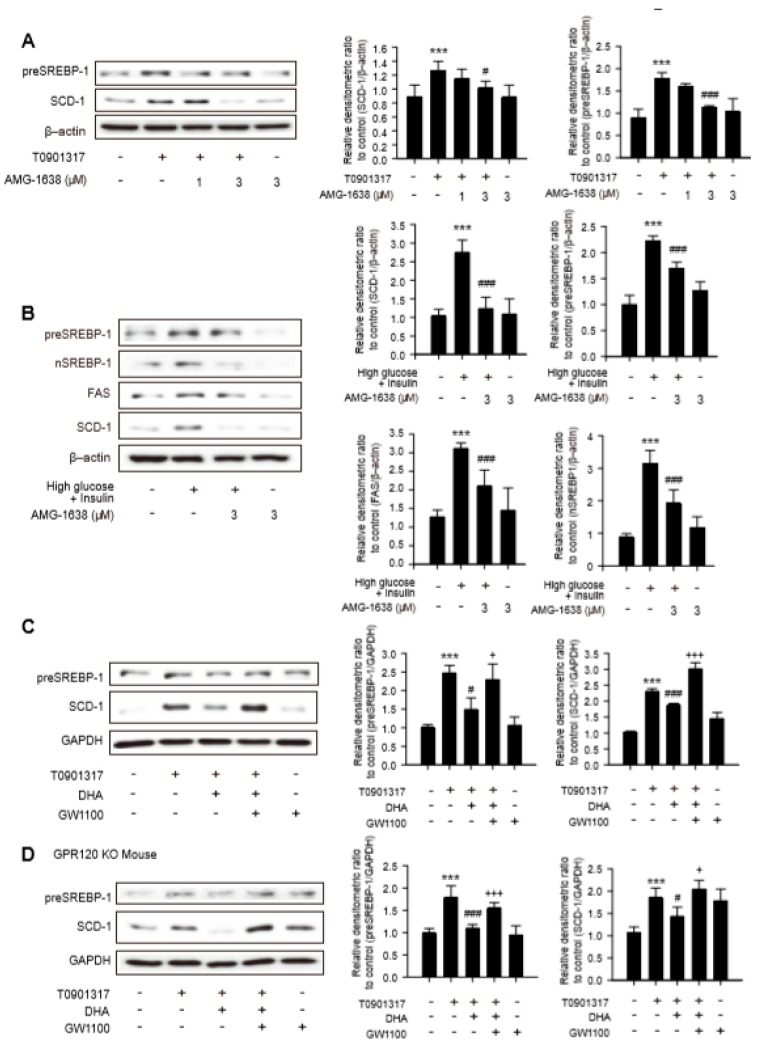
Role of GPR40 in expression of lipogenesis enzymes in hepatocytes. (**A**) Primary hepatocytes were treated with AMG-1638 (1 and 3 μM) for 12 h followed by T090 for additional 12 h, and total cell lysates were subjected to immunoblotting for preSREBP-1 and SCD-1. Data represent means ± SD (n = 4), ***P < 0.005 compared with the control group; ^#^
*p* < 0.05, ^###^
*p* <0.005 compared with the T090-treated group. (**B**) Inhibitory effects of AMG-1638 on the protein expression of lipogenic enzymes in high glucose and insulin condition. Primary hepatocytes were treated with 3 μM AMG-1638 for 12 h followed by 30 mM high-glucose medium for 30 min and further incubation with 200 nM insulin for 24 h. Data represent means ± SD (*n* = 4), *** *p* < 0.005 compared with the control group, ^###^
*p* < 0.005 compared with the T090-treated group. (**C**) Primary hepatocytes were treated with DHA and 10 μM GW1100, GPR40 antagonist for 12 h followed by 30 mM high-glucose medium for 30 min and further incubation with 200 nM insulin for 24 h. Data represent means ± SD (*n* = 4), *** *p* < 0.005 compared with the control group; # *p* < 0.05, ^###^
*p* < 0.005 compared with the T090-treated group; ^+^
*p* < 0.05, ^+++^
*p* < 0.005 compared with the T090 and DHA treated group. (**D**) In GPR120 KO hepatocytes, 10 μM GW1100 reversed the antilipogenic effects of DHA. Data represent means ± SD (*n* = 4). *** *p* < 0.005 compared with the control group; ^#^
*p* < 0.05, ^###^
*p* < 0.005 compared with the T090-treated group; +*p* < 0.05, ^+++^
*p* < 0.005 compared with the T090 and DHA treated group.

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
