# Peer review of "Involvement of G-Protein-Coupled Receptor 40 in the Inhibitory Effects of Docosahexaenoic Acid on SREBP1-Mediated Lipogenic Enzyme Expression in Primary Hepatocytes"

_ijms, 2019, doi:10.3390/ijms20112625_

Round 1

Reviewer 1 Report

This is a good experiment; however, hard to read, and its structure is strange. The authors should keep to the common line, i.e., introduction, material and methods, results,...

In addition, the results should be aggregated and ordered according to their significance. Their present a waste of data which should be aggregated to useful knowledge.

Author Response

Response to Reviewer 1 Comments

Specific comments:

(1) This is a good experiment; however, hard to read, and its structure is strange. The authors should keep to the common line, i.e., introduction, material and methods, results

: We corrected the structure in the revised manuscript.

(2) In addition, the results should be aggregated and ordered according to their significance. Their present a waste of data which should be aggregated to useful knowledge.

: We rearranged the data considering their significance. Several data were moved to supplementary figure section to aggregate the results.

Reviewer 2 Report

On et al show down-regulation of SREBP1 regulated, lipogenic enzymes in primary hepatocytes upon DHA treatment from public transcriptomic datasets, and confirmed such down-regulation in proteomic levels. They went on to show DHA treatment reduces total area of lipid droplets, but not number of droplets in primary hepatocytes upon DHA treatment. They found that such effect of DHA was mediated through GPR40, but not GPR120. 

Major points: 

(1) HepG2 is a hepatoma cell line, and whether that represents any normal hepatocytes biology is not clear. A better experimental model would be primary culture of hepatocytes in 3D organoid system

(2) How does DHA regulate GPR40 and GPR120 expressions? Author should at least discuss potential regulatory mechanisms if not showing by experiments.

Minor points:

(1) For Figure 3B, red and green channels should be presented separately. It’s difficult to see the green signal.

(2) Statistical analysis should be performed on 

Author Response

Response to Reviewer 2 Comments

Major points: 

 (1) HepG2 is a hepatoma cell line, and whether that represents any normal hepatocytes biology is not clear. A better experimental model would be primary culture of hepatocytes in 3D organoid system

: Although HepG2 is a human hepatoma cell line, it has been reported that the cell line has more similarity to normal human liver compare to other human hepatoma or hepatocyte cell lines (Huh7 and NKNT-3) in terms of hepatocyte-specific protein expression levels (Choi et al., 2015). Moreover, to represent normal hepatocyte biology, we used primary hepatocytes isolated from livers of normal C57BL/6N mice in most of the experiments. Instead of primary culture of hepatocytes in 3D organoid system, we analyzed public human microarray data to further predict the effect of DHA in normal liver.

(2) How does DHA regulate GPR40 and GPR120 expressions? Author should at least discuss potential regulatory mechanisms if not showing by experiments.

: Unlike other GPCRs that are usually downregulated by continuous stimulation with agonist, GPR40 and GPR120 were upregulated after DHA treatment in this study. This is partially consistent with the previous reports that hyperlipidemia increases the expression level of GPR40 in the islet cells (Abaraviciene, Muhammed, Amisten, Lundquist, & Salehi, 2013). Although regulatory mechanisms underlying the expression of long-chain fatty acid sensing GPCRs are not fully understood, Abaraviciene et al. showed that GPR40 protein level increased in response to 100 and 1,000 mM palmiate exposures in rat islet (Abaraviciene, Muhammed, Amisten, Lundquist, & Salehi, 2013). However, mRNA expression of GPR40 was only enhanced by 100 mM palmitate treatment. Therefore, we expect both transcriptional and post-translational regulations play a role in the enhanced expression of GPR40 and GPR120 after DHA exposure.

Minor points:

(1) For Figure 3B, red and green channels should be presented separately. It’s difficult to see the green signal.

: We presented each channel separately in the revised manuscript (figure 3B in the revised manuscript).

 (2) Statistical analysis should be performed on

: For minor review point (2), 'Statistical analysis should be performed on' is the only part that I can see (the remaining part was somehow cut off). However, I assume the comment might be about the statistical analysis on figure 2D which has no statistical error bars. We analyzed figure 2D in the original draft by summing lipid droplets in three different samples per group. Their mean values and standard errors are now reflected in the revised manuscript.

Round 2

Reviewer 1 Report

The authors have fulfilled the reviewer's recommendation and the article is ready for publication.

Author Response

Dear Editor,

           I have attached files of the revised manuscript by On et al

Response to reviewer 1 comments

Specific comments:

The authors have fulfilled the reviewer's recommendation and the article is ready for publication.

: I really appreciate your kind recommendation. Thank you.

Sincerely Yours,

 Keon Wook Kang, Ph.D.

Professor 

College of Pharmacy, Seoul National University

Gwanakro 1, Gwanak-gu

Seoul 08826, Republic of Korea

Tel: +82-2-880-7851; Fax: +82-2-872-1795; E-mail: [email protected]